# Atomic structures of anthrax toxin protective antigen channels bound to partially unfolded lethal and edema factors

Nathan J. Hardenbrook [1,4], Shiheng Liu [2,3,4], Kang Zhou [2,3,4], Koyel Ghosal [1], Z. Hong Zhou [2,3✉] & Bryan A. Krantz [1✉]

Following assembly, the anthrax protective antigen (PA) forms an oligomeric translocon that unfolds and translocates either its lethal factor (LF) or edema factor (EF) into the host cell. Here, we report the cryo-EM structures of heptameric PA channels with partially unfolded LF and EF at 4.6 and 3.1-Å resolution, respectively. The first α helix and β strand of LF and EF unfold and dock into a deep amphipathic cleft, called the α clamp, which resides at the interface of two PA monomers. The α-clamp-helix interactions exhibit structural plasticity when comparing the structures of lethal and edema toxins. EF undergoes a largescale conformational rearrangement when forming the complex with the channel. A critical loop in the PA binding interface is displaced for about 4 Å, leading to the weakening of the binding interface prior to translocation. These structures provide key insights into the molecular mechanisms of translocation-coupled protein unfolding and translocation.

[1] Department of Microbial Pathogenesis, University of Maryland, Baltimore, Baltimore, MD 21201, USA. [2] Department of Microbiology, Immunology and Molecular Genetics, University of California, Los Angeles, CA 90095, USA. [3] California NanoSystems Institute, University of California, Los Angeles, CA 90095, USA. [4] These authors contributed equally: Nathan J. Hardenbrook, Shiheng Liu, Kang Zhou. ✉email: Hong.Zhou@UCLA.edu; bkrantz@umaryland.edu

Protein translocation is an essential process in cells. Nearly one half of all proteins are translocated across a membrane to perform their respective functions[1]. This process often requires dedicated protein translocation machineries, generally referred to as translocons, to catalyze the unfolding and translocation of proteins[1]. In their native state, most proteins are thermodynamically stable. Therefore, translocons require energy in various forms, such as a proton gradient[2], hydrolysis of ATP[3], or membrane potential[3,4], to drive the translocation of their substrates. This process utilizes polypeptide clamps, or catalytic active sites that are responsible for promoting translocation of the protein. In many types of unfoldases[5], translocases[6], and secretion channels[7], these polypeptide clamps engage the polypeptide chain nonspecifically as it is unfolded and translocated. However, in the absence of high-resolution structures of translocons engaged in translocation of an unfolded protein substrate, the biophysical mechanisms involved in protein unfolding and translocation through translocons remain poorly understood.

Anthrax toxin[8] is well suited for the study of protein translocation. The toxin functions as a binary $A_2B$ toxin, with enzymatic A factors, lethal factor (LF, 91 kDa) and edema factor (EF, 89 kDa), and a cell binding B factor, protective antigen (PA, 83 kDa). Anthrax lethal factor (LF) is a 776-amino acid protein consisting of four protein domains; domain 1 is a PA-binding domain (PABD), domain 2 a VIP2-like domain, domain 3 is a helical bundle, and domain 4 is the catalytic center domain (CCD)[9]. LF has been shown to be a protease, which targets the mitogen activated protein kinase (MAPK) pathway, specifically by cleaving MAPK kinases[10–12]. Edema factor is also a four domain protein with a PA-binding domain (PABD), two adenylate cyclase domains (ACD), and a helical domain (HD)[13]. As an adenylate cyclase, EF requires calmodulin (CaM) for its activity upon translocation to the host cytosol[14]. EF has a catalytic rate of ~2000 molecules per second, resulting in high levels of cyclic adenosine monophosphate (cAMP) that activates protein kinase A (PKA) signaling pathways[15,16]. Currently, the only protein structure of EF containing all four domains is a CaM-bound crystal structure. Thus while these proteins are both translocated by PA into the cytosol, they perform very different functions.

The PA undergoes furin cleavage to form a ring-shaped homo-oligomeric pre−channel, either a heptamer or an octamer[17]. The pre-channel PA can then bind to LF or EF, forming lethal toxin (LT) or edema toxin (ET), respectively. The toxin is then endocytosed into an acidic compartment inside the cell. Within the endosomal compartment, the acidic environment induces a conformational change in the PA, resulting in the formation of a β-barrel channel that can insert itself into the endosomal membrane[18]. A proton gradient forms between the endosome and the cytosol to drive the translocation process[2]. The enzymatic factors, LF and EF, bound to the PA channel are destabilized by the acidic environment within the endosome and then unfold and translocate through the channel[19].

Atomic structures of the anthrax toxin PA pre-channel and channel have been determined by X-ray crystallography[17,20,21] and cryo-EM[18], respectively, revealing structural features supporting protein unfolding and translocation. The overall structure of the PA channel has a mushroom-shaped architecture, similar to bacterial α-hemolysin[22]. The PA channel contains three polypeptide clamp sites[23]: the α clamp[20], the ϕ clamp[24], and the charge clamp[25]. The α clamps, found on the topmost surface of PA, are clefts formed between two PA subunits that binds to α helices nonspecifically[20]. Through the α clamps and other more specific binding sites, the PA heptamers or octamers can bind three or four LF and/or EF, respectively. In addition to the α clamp, there is a binding interface formed between PA and the C-terminus of the PABD of LF (LF$_N$) in this region. PA residues K213 and K214 have been shown previously to be important in the binding of LF$_N$

to PA[26]. PA K213 was shown to interacts with D187 in LF$_N$ in this binding interface[26]. This was later shown structurally, with K213 and K214 in PA forming salt bridges with LF$_N$ residues D187 and D184, respectively[20]. A charge reversal in either of these PA residues was shown to drastically inhibit binding of LF$_N$[26]. Directly below the α clamp within the center of the channel is the ϕ clamp. The 2.9-Å resolution cryo-EM structure of the PA pore reveals that the ϕ clamp forms a constricted 6-Å bottleneck of Phe427 residues[18]. The α clamp and ϕ clamp appeared to behave in an allosteric manner that the peptide binding at the α-clamp site are required for allosteric gating of the ϕ clamp to a clamped state[27]. Below the ϕ clamp is a charge clamp formed by the transmembrane β-barrel[25]. As the partially protonated polypeptide chain moves from the lower pH in the endosome toward the higher pH within the cytosol, it passes the negatively charged acidic residues of the clamp. Here the chain becomes deprotonated, and thereafter cannot retro-translocate back through the channel. The inner diameter of the channel spans a range of diameters as low as 20 Å, wide enough to accommodate α helix in the translocating peptide, but not large enough to fit folded domains of the enzymatic factors.

Many questions remain with respect to the translocation of substrates through the PA channel. How do the catalytic domains of the substrate proteins interact with the channel? Are there binding sites beyond the α clamp that stabilize partially unfolded substrate? Are there changes within the channel structure when bound to substrate? A lack of structural information on the PA channel bound to substrate has made it difficult to address these questions. Here we report the cryo-EM structures of the PA channel bound to LF and EF. These high-resolution structures of the PA with partially unfolded protein factors reveal conformational changes occurring within the enzymatic factors upon binding to the PA channel, providing key insight on the mechanism of proton-driven protein translocation.

## Results

**Overall structures of PA channel in complex with LF and EF.** Conversion of the PA pre-channel to the channel by in vitro acidification treatment leads to rapid and irreversible aggregation due to exposure of the hydrophobic transmembrane β-barrel structure. Attempts to prevent aggregation by screening detergents were mostly unsuccessful. We next tried to apply low-pH treatment of PA pre-channels directly on carbon-coated grids as done before[17], but were only able to obtain limited number of dispersed particles of PA channel without aggregation. To overcome these issues, we used lipid nanodiscs[28] to assemble water-soluble complexes containing the PA channel bound to LF and EF[29,30]. Each complex was assembled on nickel affinity resin using His-tags in the enzyme substrates, and eluted with imidazole. The resulting complexes of PA bound by its cytotoxic substrates inserted into lipid nanodiscs provide soluble samples that take random orientation allowing for single-particle cryo-EM analysis (Supplementary Fig. 1).

The available space on the heptameric PA channel for EF binding can only accommodate up to three EF molecules due to steric hindrance. Indeed, 2D classification of cryo-EM images of PA channel bound with EF showed that the EF binding varies in different classes (Supplementary Fig. 1a, b), suggesting that the space is not fully occupied. Therefore, we used a symmetry expansion method in Relion for 3D classification and were able to resolve the asymmetrically attached EF (Methods, Supplementary Fig. 2). Remarkably, the same symmetry expansion method also worked for cryo-EM images of PA channel with LF bound even though the 2D classification failed to classify the asymmetrically bound LF in the PA-LF complex (Methods, Supplementary Figs. 1c, d and 3).

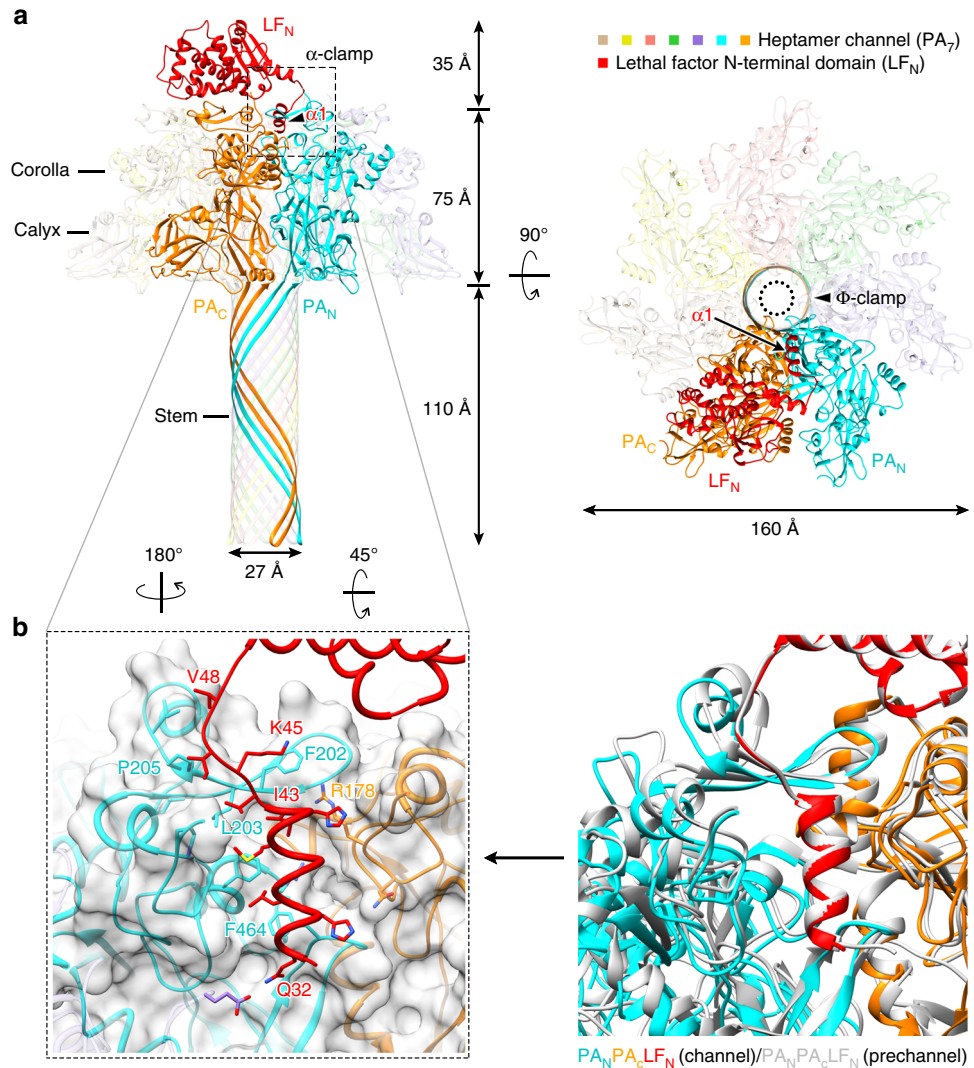

**Fig. 1 Structure of LF-bound PA₇ channel (PA₇-LF). a** Two orthogonal views of the overall PA₇-LF structure. **b** Structure comparison of substrate-binding α clamp between PA₇-LF channel and PA₈-(LF_N)₄ pre-channel.

In total, we determined four structures: one for the LF in complex with the heptameric PA channel and three for the EF(s) in complex with the heptameric PA channel, at an average resolution of 4.6 Å and 3.2–3.4 Å, respectively (Supplementary Figs. 2 and 3), based on the "gold-standard" Fourier shell correlation (FSC) 0.143 cutoff criterion[31,32]. The resulting maps revealed a "flower-on-a-stem" heptameric channel with 27-Å wide β barrel, consistent with the channel conformation. In all our structures, the conformation of the PA channel remains largely unchanged from the previously determined structure of the PA channel without substrate bound (PDB 3J9C[18]) (Figs. 1a and 2a). Atomic models of LF and EF were built into cryo-EM density maps. Only one LF is visible in the LF binding complex (refer to as PA₇-LF), while there are three configurations of the EF-bound structures: one EF and two isoforms of two EF in the EF binding complexes (referred to as PA₇-EF; PA₇-1,3-EF; PA₇-1,4-EF) (Supplementary Fig. 2). Regardless of this, we could clearly observe that an amino-terminal helix of both LF and EF binds to the α clamp of the heptameric PA channel in all complex structures; meanwhile, the rest of the amino-terminal domains of both LF and EF are well-ordered (Supplementary Fig. 4). These structures reveal how the enzymatic factors bind to the PA channel to form a complex and how the subunits in the complex interact with one another in preparation for the translocation process.

**α-clamp site from pre-channel to channel complex**. In our PA₇-LF channel structure, the LF_N (the amino terminal domain of LF) binds two neighboring PA subunits, one denoted as PA_N, which binds the N-terminus of LF_N, and the other as PA_C, which binds the C-terminus of LF (Fig. 1a). The cryo-EM density reveals a helix of LF_N bound in the α-clamp site (Supplementary Fig. 4), indicating that this site continues to engage the enzymatic factors in the PA channel. This helix of LF_N in PA₇ α-clamp appears to bind within this site in much the same manner as in the PA₈-(LF_N)₄ pre-channel structure (PDB 3KWV[20]) (Fig. 1b). However, at 4.6-Å resolution, it is not possible to determine whether hydrogen bonds form within the α-clamp site between LF_N β1 and PA_N β13. Upon alignment with 3KWV[20], nonetheless, β13 in the PA channel and in the PA₈-(LF_N)₄ pre-channel has a highly similar conformation, with the first α-helix (LF_N α1) aligning well between the two structures. This alignment indicates that LF_N α1 binds within the α-clamp site similarly in both the PA channel and the pre-channel. The catalytic domain of LF is invisible in our EM structure, suggesting that it is flexible.

The amino terminal domain of EF (EF_N) and LF_N share similar structures upon binding to the PA channel (Figs. 2a and 1a). In the crystal structures free of PA binding, LF α1 is an ordered α-helix[9] but the homologous region in EF is flexible and disordered[33]. Upon PA binding, these disordered residues of EF

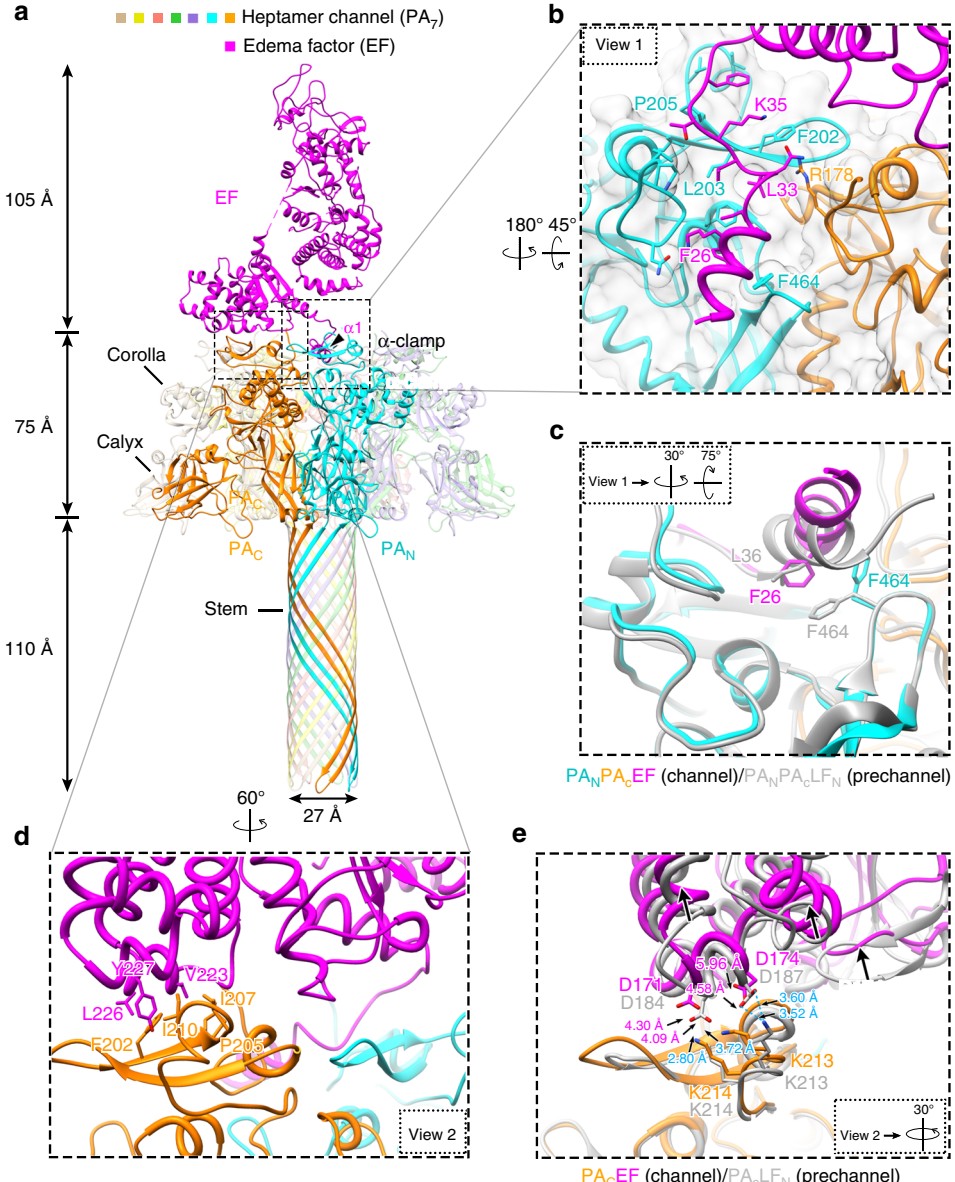

**Fig. 2 Structure of EF-bound PA$_7$ channel (PA$_7$-EF). a** Overall structure of PA$_7$-EF shown as ribbon. **b** Zoom-in view (view 1) of the PA$_7$ α-clamp site showing its detailed interactions with α1 of EF. The cryo-EM density is shown as semi-transparent gray. **c** Rotated view 1 showing structure comparison of the substrate-binding α-clamp between PA$_7$-EF channel (color) and PA$_8$-(LF$_N$)$_4$ pre-channel (gray), except that the density is not shown for clarity. **d** Zoom-in view (view 2) showing the details of the PABD domain of EF binding to PA$_N$ and PA$_C$. **e** Rotated view 2 showing the superposition of PA-bound EF (purple for EF, orange for PA) and PA-bound LF (gray for both PA and LF), except that the density is not shown for clarity. Hydrogen bonds are shown as dashed lines.

(residues 20–30) refold into an α-helix (EF$_N$ α1) and bind within the α-clamp site (Fig. 2b). β1 (Leu33 to Lys35) of EF$_N$ forms parallel β-sheet with β13 (Leu203 to Pro205) of PA$_N$ (Fig. 2b). The hydrogen bonds between the two β-strands are analogous to those found in the PA$_8$-(LF$_N$)$_4$ pre-channel structure (PDB 3KWV[20]), confirming predictions that the amino terminus of EF binds in a similar way as LF[20].

**Plasticity of helix binding within the α-clamp site.** While α1 and β1 in LF and EF bind to the α clamp of PA analogously with β1 forming hydrogen bonds with PA$_N$, their α1 helices dock within the α clamp differently, indicating that there is structural plasticity of α-helix binding within this α-clamp site (Fig. 2c). LF α1 is angled downward towards the pore in the α-clamp site,

while the amino-terminal end of EF α1 is elevated ~2.9 Å as measured using the carbonyl groups on LF Glu34 and EF Glu24. This elevation in EF α1 appears to be caused by a change in the orientation of PA$_N$ Phe464. The phenyl ring in PA$_N$ Phe464 is positioned outwards toward the bound EF α1. This change in the orientation of Phe464 appears to restrict EF α1 in its elevated conformation in the α-clamp site. Overall, this structural plasticity makes sense, given previous work determining that the α clamp in PA binds α helices repeatedly and nonspecifically during translocation of its substrates[27].

**Interface destabilization may play role in translocation.** A hydrophobic interface is formed in the carboxy-terminal subdomain of EF$_N$ with PA between EF residues Val223, Leu226,

Tyr227 and $PA_C$ residues Phe202, Pro205, Ile207, Ile210 (Fig. 2d). Like those in the $PA_8$-$(LF_N)_4$ pre-channel complex crystal structure previously determined[20], $PA_C$ Ile210 and EF Tyr227 are well packed in this hydrophobic interface, allowing the phenol hydroxyl to form hydrogen bonds within the interface with $PA_C$ residues His211 and Asp195 (Fig. 2d).

Despite the above similarity, it is worth noting that, upon undergoing the conformational change from pre-channel to channel, the substrate appears to have moved up (Fig. 2e), away from the binding interface with $PA_C$ compared to the pre-channel structure[20]. This conformational change has resulted in a loss of salt bridges that had previously formed upon binding of the LF to the pre-channel between residues $PA_C$ Lys213 and Lys214 and LF Asp187 and Asp184, respectively, where the distance between $PA_C$ Lys213 and LF Asp187 increases from 3.5 Å in the pre-channel structure to 4.6 Å in the $PA_7$-EF channel structure; the distance between $PA_C$ Lys214 and LF Asp184 increases from 2.8 Å in the pre-channel structure to 4.3 Å in the $PA_7$-EF channel structure (Fig. 2e). The loss of these salt bridges in the binding interface of $PA_7$-LF should destabilize the binding interface, preparing the substrate for subsequent dissociation and unfolding prior to its translocation. Indeed, when we mutate EF residues Asp171 and Asp174 to alanine, we see no change in binding affinity compared to wild type using planar bilayer electrophysiology (Supplementary Fig. 6). This result indicates that the salt bridges are weakened significantly once PA reaches the channel state. Thus, the LF/EF binding interface with PA can be maintained in a higher affinity mode when PA is in the pre-channel conformation and complex assembly is more important; but when PA converts into the channel state, the affinity of the LF/EF binding interface is destabilized, allowing for more rapid dissociation and unfolding of LF/EF during translocation.

**EF domains reorganize upon binding the PA channel**. Unlike LF, all the domains of EF are well resolved in our channel complex structures (Fig. 3a, b). The corresponding amino acid sequence for the different domains with respective α helices and β sheets is shown in Fig. 3c. In all our structures of the EF-bound channel ($PA_7$-EF, $PA_7$-1,3-EF, and $PA_7$-1,4-EF) presented here, EF undergoes a significant conformation change compared to its calmodulin (CaM)-bound structure[33]. In the previous CaM-bound EF structure (PDB: 1XFY[33]), CaM stabilizes the $C_A$ and $C_B$ ACD and the HD of EF, and there are no significant interactions among these domains of EF (Fig. 3e). While in the $PA_7$-bound EF structures, the HD domain contributes to a new conformation by bridging the PABD and ACD. Further analysis indicated that the folding pattern within the three domains only changes slightly from CaM-bound EF to $PA_7$-bound EF, but the three domains are reorganized in $PA_7$-bound EF (Fig. 3e and Supplementary Movie 1).

In more detail, on one side of the HD, residues near α29 and α30 of HD interact with those near α2 and β1 of the PABD. Hydrogen bonds are formed between inter-domain residues, one from HD and the other from PABD, such as Gln746-Asn40, Lys767-Gln50, Asn737-Ile71, Asn737-Phe73, and Glu739-Phe73 (Fig. 3d). On the other side of HD, a loop between α26 and α27 interacts with residues near α22 and α24 of ACD, mainly through inter-domain hydrogen bonds (Fig. 3d). With the extensive interactions mentioned above, HD moves toward and binds PABD eventually. Notably, the refolding of N-terminal residues (Lys20 to Thr42) of PABD (Fig. 3e), which is a consequence of $PA_7$ binding, yields the space that enables the interactions between HD and PABD (Fig. 3d, pink arrow in Fig. 3e, and Supplementary Movie 1), leading to a 60° swing of ACD (yellow arrow in Fig. 3e), which mounts α22 and α24 of ACD on the loops near α25, α26, and α27 of HD (Fig. 3d).

## Discussion

Here we report a total of four cryo-EM structures of heptameric PA channel bound with toxin substrates: three for the complex with EF at resolutions ranging from 3.2 to 3.4 Å and one for the complex with LF at 4.6 Å resolution. Our results reveal that upon the binding of the substrate to the PA channel, conformational changes occur in the enzymatic substrates LF and EF. When full-length LF binds to the PA channel, its catalytic domains exhibit significant flexibility, and thus only the PA binding domain, $LF_N$, is visible in the cryo-EM density map; by contrast, the PA-binding and catalytic domains are visible in the crystal structure of LF[9]. In the case of EF, its domains reorganize, compared to the EF structure bound to CaM[33]. This CaM-bound structure is the only other full-length structure of EF available for comparison, but highlights the conformational changes EF undergoes during its lifetime. It is interesting that EF binds to PA in a different way than it binds to CaM, with the helical domain stabilizing its PABC and ACD. This reorganization involves refolding of PABD residues (Lys20 to Thr42), a 70° swing of HD toward PABD and mounting of ACD to HD. The reorganized conformation of EF is stabilized by the formation of hydrogen bonds. We suggest that this reorganization of the domains plays a role in the ordered translocation of the EF through the channel. Previously, Feld et al. showed in detail that different substrates could bind to the α clamp[20], indicating nonspecific binding at the α-clamp site. Our results also show the α clamp engages different α helices, either from EF or LF. Interestingly, PA's Phe464, a residue lining the α clamp, changes conformation to accommodate different residues in these helices. These results demonstrate plasticity within the α-clamp site, which allows for the binding of different helical substrates. When bound to the pre-channel, $LF_N$ forms numerous stabilizing interactions on its amino and carboxyl terminal sub-domains. Upon conversion to the channel conformation, the carboxyl terminal subdomain of $LF_N$ destabilizes its interface with PA. This destabilization occurs while the complex is exposed to the acidic pH of the endosomal compartment. This interface destabilization, paired with the acidic environment, most likely plays an important role in allowing the bound substrate to unfold and translocate through the channel more efficiently.

Our four high-resolution structures of PA channel with LF and EF—representing the structures of the complex in the channel conformation—provide further insights into the mechanism of how substrate proteins are translocated across membranes by the PA channel. In our current model (Fig. 4), the enzymatic factors bind to PA pre-channels, before the cell undergoes endocytosis. The PA prechannel undergoes a conformational change within the endosomal compartment, forming the channel state. This conformational change results in an alteration of the binding interaction between the channel and its substrate enzymes, thus destabilizing the interaction. This destabilization, accompanied by partial protonation of the polypeptides, allows the proton gradient to drive translocation of the bound substrate through the channel. As the polypeptide is translocated through the channel, it is engaged by the α-clamp repeatedly and non-specifically[27]. During much of the translocation process, the polypeptide is accommodated by the channel in its secondary structure. It is engaged as an α helix while binding within the α clamp. As it moves down and is bound in the φ-clamp site, the α-clamp engages the polypeptide again. When the α clamp re-engages with the polypeptide, it causes an allosteric change in the φ clamp[24]. This change in the φ clamp applies force to the α-helix, changing

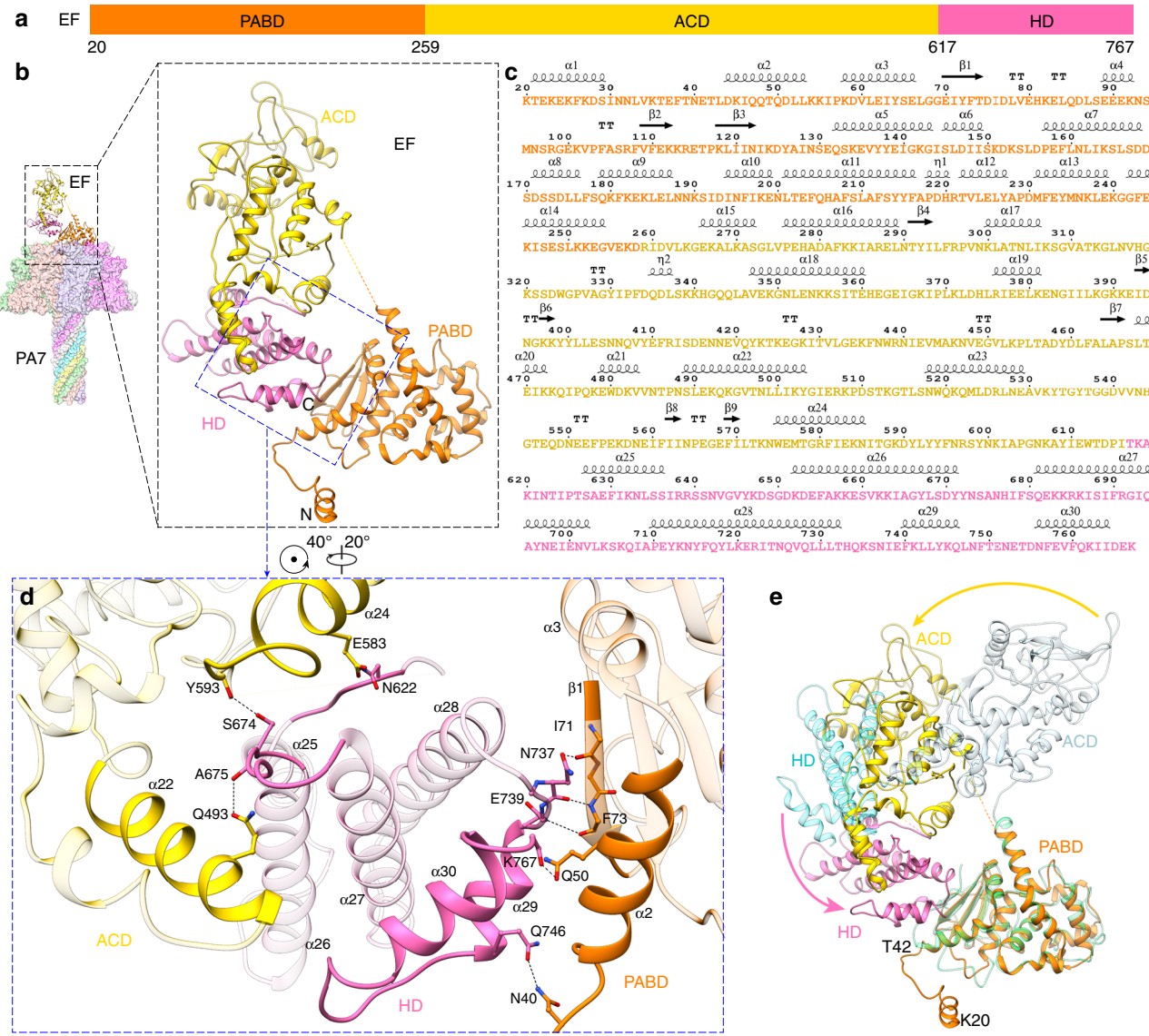

**Fig. 3 Structural comparison of EF between its PA7-bound and CaM-bound forms. a** Domain architecture of EF with individual domains colored and the boundary residues numbered. **b** Structure of PA7-EF with EF shown as ribbon and PA7 as surface colored by protomers. The three domains of EF—PABD, ACD, HD—are colored as in **a**. **c** Sequence and secondary structures of the PA7-bound EF. **d** Close-up view at the interactions among PABD, ACD and HD domains in PA7-bound EF. The structural elements involved in domain interactions are highlighted and hydrogen bonds are shown as dashed lines. **e** Superposition of EF structures in its PA7-bound and CaM-bound (PDB: 1XFY) forms. The two EFs are aligned by the PABD domain for clarity. Three domains of CaM-bound EF—PABD, ACD, HD—are colored in green, light blue and cyan, respectively. Domain reorganizations are marked by arrows.

its conformation to extended chain and driving it past the charge clamp site. Once past the ϕ clamp, the polypeptide is deprotonated within the anionic charge clamp. This prevents retro-translocation of the polypeptide chain back toward the endosome. At this point the polypeptide can begin to reform its secondary structure. Once exiting the channel, the translocating polypeptide refolds into its tertiary structure and can perform its enzymatic effects on the host cytosol. In the case of EF, this involves binding CaM and taking on its CaM-bound domain organization[33].

Recently, structures of the substrate-engaged SecY protein translocon have been determined using X-ray crystallography and cryoEM[34,35]. The SecY system is one of the few other protein translocation systems where structural information is available. Like PA, within the SecY channel there is a hydrophobic pore ring that interacts with the translocating polypeptide. In addition, a polypeptide clamp has been identified in SecA which would position the translocating polypeptide right above the SecY

pore[36]. The recent structure of the clamp bound to the translocating substrate indicates that it engages with the polypeptide in a sequence-independent manner by inducing short β strand conformations in the polypeptide[35]. This action would allow a broad range of polypeptides to be bound and translocated by the SecA. Hence this clamp is like the α clamp in PA, which also engages multiple sequences. This similarity suggests that perhaps there are universally shared phenomenon amongst different translocons, in which substrate is engaged sequence-independently based on secondary structure. In general, these two translocons allow different polypeptide segments to be engaged repeatedly and non-specifically as they translocate through their respective channels.

## Methods

**Protein expression and purification.** Heptameric PA oligomer (PA7) was prepared as described[17]. Briefly, PA83 was expressed in *Escherichia coli* BL21(DE3) using a pET22b plasmid directing expression to the periplasm. PA83 was extracted

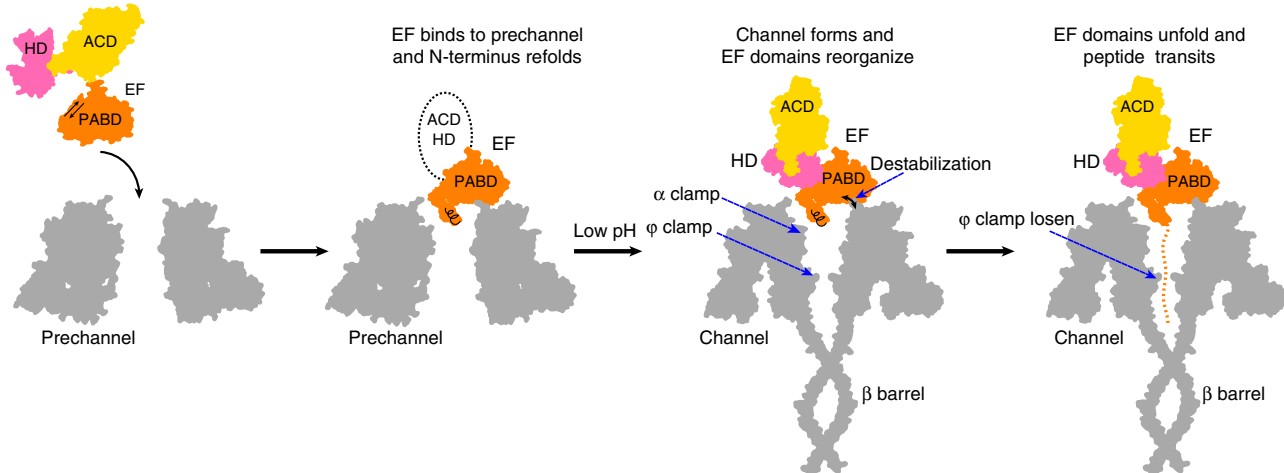

**Fig. 4 Mechanism of EF translocation.** Illustration of the anthrax toxin channel translocation steps with EF. Initially, EF binds to the PA pre-channel, and the N-terminal α helix of the PABD of EF docks into the α clamp, yielding the space for domain reorganization of EF. After the PA pre-channel changes to the channel state at low pH, the destabilization of the interface between the PABD of EF and the PA channel allows the N-terminal α helix to translocate down to the φ-clamp site. In parallel, the α clamp engages the EF polypeptide again, causing an allosteric change in the φ clamp. The change in the φ clamp applies force to the α helix, changing its conformation to extended chain and driving it past the charge clamp site located near the top of the β barrel. The cycle repeats on the next section of EF polypeptide.

from the periplasm and further purified using Q-Sepharose anion-exchange chromatography in 20 mM Tris-chloride, pH 8.0, and eluted with a gradient of 20 mM Tris-chloride, pH 8.0 with 1 M NaCl. $PA_{83}$ was then treated with trypsin (1:1000 wt/wt trypsin:PA) for 30 min at room temperature to form $PA_{63}$. The trypsin was inhibited with soybean trypsin inhibitor at 1:100 dilution (wt/wt soybean trypsin inhibitor:PA). The trysinized PA was subjected to anion-exchange chromatography to isolate the oligomerized $PA_7$. The trypsinized PA was applied to the anion exchange column in 20 mM Tris-chloride, pH 8.0, and the oligomerized $PA_7$ was eluted from the anion exchange column using a gradient of 20 mM Tris-chloride, 1 M sodium chloride, pH 8.0. Recombinant WT LF and WT EF and EF point mutants, containing an amino-terminal six-histidine His-tag ($His_6$) were overexpressed in *Escherichia coli* BL21(DE3) from pET15b constructs and purified from the cytosol using $His_6$ affinity chromatography. Cytoplasmic lysates of $His_6$-LF and $His_6$-EF were made by treatment with hen egg white lysozyme for 30 min at room temperature. The lysates were briefly sonicated at 4 °C (for 2 min) to shear genomic DNA and reduce sample turbidity. $His_6$-LF and $His_6$-EF lysates were applied to immobilized nickel affinity chromatography column in 20 mM Tris-chloride, 35 mM imidazole, 1 M sodium chloride pH 8.0, and $His_6$-LF and $His_6$-EF were eluted using a gradient of 20 mM Tris-chloride, 500 mM imidazole, 1 M sodium chloride pH 8.0. Affinity-purified $His_6$-LF and $His_6$-EF were then subjected to S200 gel filtration chromatography in 20 mM Tris-chloride, 150 mM sodium chloride, pH 8.0. EF point mutants were made using the Quik-Change mutagenesis kit (Stratagene) according to the manufacturers procedure with the primer designs listed in Supplementary Table 1.

**PA-LF and PA-EF complex assembly.** $His_6$-LF or $His_6$-EF were mixed with $PA_7$ pre-channel at a ratio of 5:1 (LF/EF:$PA_7$) and allowed to assemble on ice for 1 h. The $PA_7$ pre-channel in complex with $His_6$-LF and $His_6$-EF was then purified over S400 gel filtration in 20 mM Tris-chloride pH 8.0, 150 mM sodium chloride.

**Nanodisc insertion.** The $His_6$ tag was removed from membrane scaffold protein 1D1 (MSP1D1)[28]. pMSP1D1 was a gift from Stephen Sligar (Addgene plasmid #20061). In all, 300 μL wet volume Ni-NTA Superflow resin (Qiagen) was added to an 800-μL centrifuge column (Pierce) twice with 50 mM sodium chloride, 50 mM Tris-chloride pH 7.5 (Buffer A). In all, 300 μL of 1 μM of our PA complex and 300 μL of 2 M urea were added to the resin, for a final urea concentration of 1 M. This mix was then collected and incubated at 37 °C for 5 min to induce conversion from the pre-channel to channel conformation[29]. The mix was then collected and added back into a centrifuge column, and the resin (now bound to complex) was washed twice with 500 μL Buffer A. A mixture containing MSP1D1 and palmitoyloleoyl phosphocholine (POPC) was made by first evaporating chloroform off of POPC, then adding MSP1D1 and sodium cholate in Buffer A. The final concentration contained 4 μM MSP1D1, 400 μM POPC, and 25 mM sodium cholate in Buffer A. In all, 500 μL of a MSP1D1-(POPC) mix was added to the dry resin bound with PA complex[30]. This resin slurry was then collected and dialyzed in Slide-A-Lyzer cassette (10 kDa molecular weight cut-off) (Thermo Scientific) in excess Buffer A for 8–12 h at a time, with two buffer changes. The Ni-NTA was then collected after dialysis. The resin was washed twice with 500 μL Buffer A. The resin was then washed with 500 μL of 50 mM NaCl, 50 mM imidazole, 50 mM Tris pH 7.5 to

elute any remaining proteins bound non-specifically. The nanodisc complex was then eluted in 50 mM sodium chloride, 300 mM imidazole, 50 mM Tris-chloride pH 7.5. This eluted sample was then dialyzed into Buffer A and concentrated to 0.274 mg ml$^{-1}$ (PA channel in complex with LF) and 0.498 mg ml$^{-1}$ (PA channel in complex with EF) Concentration was estimated by a Nanodrop spectrophotometer.

**Cryo-EM sample preparation and imaging.** For cryo-EM sample optimization, an aliquot of 2.5 μl of sample was applied onto a glow-discharged holey carbon copper grid (300 mesh, QUANTIFOIL® R 2/1). The grid was blotted and flash-frozen in liquid ethane with an FEI Mark IV Vitrobot. An FEI TF20 cryo-EM instrument was used to screen grids. Cryo-EM grids with optimal particle distribution and ice thickness were obtained by varying the gas source (air or $H_2/O_2$), time for glow discharge, the volume of applied samples, chamber temperature/humidity, blotting time/force. For the PA channel in complex with LF, our best grids were obtained using $H_2/O_2$ for glow discharge and with the Vitrobot sample chamber set at 12 °C temperature and 100% humidity. For the PA channel in complex with EF, our best grids were obtained using air for glow discharge and with the Vitrobot sample chamber set at 16 °C temperature and 100% humidity.

Optimized cryo-EM grids were loaded into an FEI Titan Krios electron microscope with a Gatan Imaging Filter (GIF) Quantum LS device and a post-GIF K2 Summit direct electron detector. The microscope was operated at 300 kV with the GIF energy-filtering slit width set at 20 eV. Movies were acquired with Leginon[37] by electron counting in super-resolution mode at a pixel size of 0.535 Å per pixel. A total number of 45 frames were acquired in 9 s for each movie, giving a total dose of ~60 e$^-$/Å$^2$/movie.

**Drift correction for movie frames.** Frames in each movie were aligned for drift correction with the graphics processing unit (GPU)-accelerated program MotionCor2[38]. The first frame was skipped during drift correction due to concern of more severe drift/charging of this frame. Two averaged micrographs, one with dose weighting and the other without dose weighting, were generated for each movie after drift correction. The averaged micrographs have a calibrated pixel size of 1.07 Å on the specimen scale. The averaged micrographs without dose weighting were used only for defocus determination and the averaged micrographs with dose weighting were used for all other steps of image processing.

**Structure determination for PA channel in complex with EF.** For the PA channel in complex with EF, the defocus value of each averaged micrograph was determined by CTFFIND4[39] generating values ranging from −1.5 to −3 μm. Initially, a total of 1,481,285 particles were automatically picked from 6811 averaged images without reference using Gautomatch (http://www.mrc-lmb.cam.ac.uk/kzhang). The particles were boxed out in dimensions of 256 × 256 square pixels square before further processing by the GPU accelerated RELION2.1. Several iterations of reference-free 2D classification were subsequently performed to remove bad particles (i.e., classes with fuzzy or un-interpretable features), yielding 725,251 good particles. The reported map of the heptameric anthrax toxin PA channel[18] (EMD-6224) was low-pass filtered to 60 Å to serve as an initial model for 3D classification. After one round of 3D classification with C7 symmetry, only the

classes showing feature corresponding to the intact $PA_7$ channel were kept, which contained 486,169 particles. We re-centered those particles and removed duplications based on the unique index of each particle given by RELION[32]. The resulting 486,169 particles were applied one round of auto-refinement by RELION, yielding a map with an average resolution of 3.0 Å.

Next, we expanded C7 symmetry to C1, yielding 3,403,183 (486,169 × 7) particles. These particles were submitted to further classification (skip align) with 29 classes. A cylinder mask was created only for the EF binding region (Supplementary Fig. 2) and applied for the focus classification. Among these 29 classes, four different types of density maps were identified. Four classes have no clear density of EF ($PA_7$), 14 classes show clear density of only one EF binds to the $PA_7$ channel ($PA_7$-EF), six classes with density of two EF ($PA_7$-1,3-EF), and four classes with density of two EF which were located further away from each other ($PA_7$-1,4-EF) (Supplementary Fig. 2). Subsequently, we merged the particles from classes belonging to $PA_7$-EF, $PA_7$-1,3-EF, $PA_7$-1,4-EF, respectively. After removing duplications based on the unique particle names given by RELION, we got 333,455 particles for $PA_7$-EF (68.8% of all particles), 72,864 particles for $PA_7$-1,3-EF (15.0% of all particles) and 73,784 particles for $PA_7$-1,4-EF (15.1% of all particles).

The unique particles of each dataset ($PA_7$-EF, $PA_7$-1,3-EF, $PA_7$-1,4-EF) resulting from the focused classification were subjected to a final step of 3D auto-refinement with C1 symmetry. The two half maps of each dataset from this auto-refinement step were subjected to RELION's standard post-processing procedure. The final maps of $PA_7$-EF, $PA_7$-1,3-EF, $PA_7$-1,4-EF achieved an average resolution of 3.2, 3.4, and 3.4 Å, respectively, based on RELION's gold-standard FSC (see below).

**Structure determination for PA channel in complex with LF**. For the PA channel in complex with LF, the defocus value of each averaged micrograph was determined by CTFFIND4[39] generating values ranging from −1.5 to −3 μm. Initially, a total of 616,153 particles were automatically picked from 2502 averaged images without reference using Gautomatch (http://www.mrc-lmb.cam.ac.uk/kzhang). The particles were boxed out in dimensions of 320 × 320 square pixels square and binned to 160 × 160 square pixels (pixel size of 2.14 Å) before further processing by the GPU accelerated RELION2.1. Several iterations of reference-free 2D classification were subsequently performed to remove bad particles (i.e., classes with fuzzy or un-interpretable features), yielding 204,395 good particles. The reported map of the heptameric anthrax toxin PA channel[18] (EMD-6224) was low-pass filtered to 60 Å to serve as an initial model for 3D classification. After one round of 3D classification with C7 symmetry, only the classes showing feature corresponding to the intact $PA_7$ channel were kept, which contained 194,849 particles. We recentered those particles and removed duplications based on the unique index of each particle given by RELION. The resulting 194,775 particles were un-binned to 320 × 320 square pixels (pixel size of 1.07 Å) and applied one round of auto-refinement by RELION, yielding a map with an average resolution of 3.4 Å.

The C7 symmetry was then expanded to C1, giving 1,363,425 (194,775 × 7) particles for further classification. A cylinder mask was created only for the LF-binding region (Supplementary Fig. 3) and applied for the focus classification with seven classes. Six of the seven classes show clear density for only one LF bound to the $PA_7$ channel ($PA_7$-LF) (Supplementary Fig. 3). We next merged the good particles from the six classes and removed duplications based on the unique particle names given by RELION.

The 63,807 un-binned, unique particles (10.4% of all particles) resulting from the focused classification were subjected to a final step of 3D auto-refinement with C1 symmetry. The two half maps from this auto-refinement step were subjected to RELION's standard post-processing procedure. The final map of the $PA_7$-LF complex has an average resolution of 4.6 Å based on RELION's gold-standard FSC. We also got a 3D auto-refinement result (3.6 Å) with C7 symmetry using this dataset, which helped the model building process (see model building below).

**Resolution assessment**. All resolutions reported above are based on the "gold-standard" FSC 0.143 criterion[40]. FSC curves were calculated using soft spherical masks and high-resolution noise substitution was used to correct for convolution effects of the masks on the FSC curves[41]. Prior to visualization, all maps were sharpened by applying a negative B-factor, which was estimated using automated procedures[20].

Local resolution was estimated using ResMap[42]. The overall quality of the maps for the PA channel in complex with EF and LF is presented in Supplementary Figs. 2 and 3, respectively. Data collection and reconstruction statistics are presented in Supplementary Table 2.

**Model building and refinement**. Atomic model building was accomplished in an iterative process involving Coot[43], Chimera[44], and Phenix[45]. For the $PA_7$-LF complex, the structure of anthrax toxin PA channel heptamer[18] (PDB ID: 3J9C) was fitted into cryo-EM map (4.6 Å, C1 symmetry) by using the 'fit in map' routine in Chimera. The atomic model building of $PA_7$ channel was facilitated by using the 3.6 Å cryo-EM map in C7 symmetry (63,807 particles, Supplementary Fig. 2). Next, the crystal structure of LF[9] (PDB ID: 1J7N) was fitted in to the cryo-EM map (4.6 Å, C1 symmetry) to create a full atomic model for $PA_7$-LF. Finally, the structure was manually adjusted using Coot and refined using Phenix in real space with secondary structure and geometry restraints.

For the PA channel in complex with EF, we have three different types of density maps—$PA_7$-EF, $PA_7$-1,3-EF, and $PA_7$-1,4-EF. Owing to the higher resolution and single EF binding in $PA_7$-EF, we firstly carried out model building on this density map. The structure of $PA_7$ channel[18] (PDB ID: 3J9C), was fitted into the cryo-EM map of $PA_7$-EF as initial model by using the 'fit in map' routine in Chimera. This fit revealed the extra density corresponding to EF. However, further docking showed the density of EF in cryo-EM map has significant differences with respect to the crystal structure of EF[33] (PDB ID: 1XFX). The full-length EF consists of four domains, the (PABD), two catalytic core domains $C_A$ and $C_B$ forming the ACD, and the HD. The domains in the cryo-EM map have a different arrangement, however. Thus, we fit the domains into the density separately to create an initial atomic model for $PA_7$-EF, which was refined by "real-space refinement" in Phenix. We then manually adjusted the main chain and side chains to match the cryo-EM density map with Coot. This process of real space refinement and manual adjustment steps was repeated until the peptide backbone and side chain conformations were optimized. Secondary structure and geometry restraints were used during the refinement.

Refinement statistics of the PA channel in complex with LF and EF are summarized in Supplementary Table 2. These models were also evaluated based on MolProbity scores[46] and Ramachandran plots (Supplementary Table 2). Representative densities for the proteins are shown in Supplementary Fig. 4.

**Planar lipid bilayer electrophysiology apparatus**. Planar lipid bilayer currents were recorded using an Axopatch 200B amplifier and a Digidata 1440 A acquisition system (Molecular Devices Corp., Sunnyvale, CA)[17,47]. Ensemble recordings were recorded at 200 Hz and filtered at 100 Hz using PCLAMP10 software. The membrane potential difference is defined as $\Delta\psi \equiv \psi_{cis} - \psi_{trans}$ ($\psi_{trans} \equiv 0$ V).

**Ensemble binding analysis using electrophysiology**. A prior method[20] was used to monitor EF binding to PA channels at symmetrical pH and a $\Delta\psi$ of 0 mV by means of an applied potassium chloride gradient. Membranes were painted on a 100 μm aperture of a 1-mL, white-Delrin cup with 3% (wt/vol) 1,2-diphytanoyl-$sn$-glycerol-3-phosphocholine (Avanti Polar Lipids, Alabaster, AL) in $n$-decane (Sigma-Aldrich, St. Louis, MO); and the cis chamber was bathed in 10 mM potassium phosphate, 100 mM potassium chloride, pH 7. During the setup, the trans chamber was bathed in 10 mM potassium phosphate, pH 7. PA channels were inserted by adding 20 nmol of $PA_7$ to the cis chamber at pH 7. PA currents reached ~5 nA. Upon stabilization of the ensemble current, the cis chamber was perfused to exchange fresh 10 mM phosphate, 100 mM KCl at pH 7. EF and mutants thereof were added in small increments to the cis side of the membrane, allowing for binding to reach equilibrium as indicated by the observed decrease in current which reached a steady-state plateau. Fraction of closed channels ($\theta_{obs}$) versus [P] plots (where P denotes free EF) were fit to a simple single-site model, $\theta_{obs} = 1/(1 + K_D/[P])$, to obtain an equilibrium dissociation constant, $K_D$. Three to four independent measurements of $K_D$ for each EF mutant and wild type were made and averages and standard deviations were computed in Microcal ORIGIN9 software.

**Reporting summary**. Further information on research design is available in the Nature Research Reporting Summary linked to this article.

## Data availability

The cryo-EM maps have been deposited in the Electron Microscopy Data Bank under accession numbers EMD-20459, EMD-20955, EMD-20957, and EMD-20958. The atomic structure coordinates have been deposited in the Protein Data Bank under the accession numbers 6PSN [https://doi.org/10.2210/pdb6PSN/pdb], 6UZB [https://doi.org/10.2210/pdb6UZB/pdb], 6UZD [https://doi.org/10.2210/pdb6UZD/pdb], and 6UZE [https://doi.org/10.2210/pdb6UZE/pdb]. The source data underlying Supplementary Fig. 6 are provided as a Source Data file. Other data can be obtained from the corresponding authors upon reasonable request.

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

## Acknowledgements

We thank J. Jiang for their suggestions about sample preparation and data processing, Y. Cui for assistance in cryo-EM and suggestions about data processing, I. Atanasov and W. Hui for assistance in cryo-EM. This work was supported in part grants from the National Science Foundation (NSF, under grant no. DMR-1548924) and by grants from the National Institutes of Health (R01GM071940/AI094386/DE025567 to Z.H.Z. and R21AI124020 to B.K.) and the Training Program in Integrative Membrane Biology at the University of Maryland, Baltimore (T32GM008181). We acknowledge the use of resources in the Electron Imaging Center for Nanomachines supported by UCLA and grants from the NIH (S10RR23057, S10OD018111, and U24GM116792) and NSF (DBI-1338135). K.Z. acknowledges support from the China Scholarship Council.

## Author contributions

Z.H.Z. and B.K. conceived the project; N.J.H. engineered and isolated samples; S.L. and K.Z. evaluated the samples, performed electron microscopy, processed the data, built atomic models, and prepared figures; N.J.H and K.G. performed equilibrium binding electrophysiology experiments; all authors wrote the paper.

## Competing interests

The authors declare no competing interests.
