## [Peer Review File · Nature Communications]

Reviewers' comments:

Reviewer #1 (Remarks to the Author):

In this manuscript, Hardenbrook and co-authors, presents two models of the anthrax toxin complexes with the protective antigen (PA) in a pore conformation: one bound to the lethal factor (LF) and the other to the edema factor (EF). The models were build using two cryo-EM maps, one with one N-terminal domain of LF (LFN) and the other with one EF molecule bound at the surface of the PA pore. The authors also obtained two other maps with EF that contain two EF molecules at different position at the surface of the PA heptameric pore (which they did not analyze). Novelities of the work are the fact that PA is in a pore conformation and that this paper presents the first structure of PA bound to EF. Not surprisingly, given the similarity of the interacting surface between the PA oligomer in the pore and prepore conformations, LFN is seen interacting in a similar manner to that observed by X-ray crystallography and published by the same group (Feld et al, 2010), so this is not particularly novel. Most interesting is the structure of PA-EF, although as explained below, the authors should be careful in their interpretation.

The quality of cryo-EM work is of high standard and the use of symmetry expansion combined with masking for classification smartly done. However, the stoichiometry of binding obtained differs from previous report of the literature, where it has been clearly established that three substrates molecules can bind at the surface of PA heptamer. In this work, the authors pre-assembled and purified PA-LF and PA-EF complexes, then converted the PA to a pore conformation using urea (2M). The reduction in the number of substrates after urea treatment is highly concerning as is the unfolding of LFC. The concern here is that urea also unfolds LF (Lo et al, 2015). It is also to be noted that one remaining LFN (instead of three full-length molecules) is seen in 10% of the complexes (or after losing 97% of the bound substrate). What is the effect of urea on EF? Careful biochemical characterization of the substrates and the complexes should be conducted before being able to reach the conclusions made regarding the reorganization of the substrates upon binding to the PA channel ("EF domains reorganize upon binding the PA channel"). Could the domains reorganization of EF described Fig 3 be due to the urea treatment? A better approach would be to convert PA to a pore before assembling the complex, this way avoiding exposing LF and EF to urea.

An important concern as a reviewer is that the authors focus on the models and do not present the experimental evidence that support the models, especially regarding the position of N-terminal helix of EF at the surface of PA. The authors state that the N-terminal helix of EF interacts with the α -clamp at the surface of PA. But the authors only show the model (Fig 2): they authors do not show the corresponding densities in any of three EM maps of PA-EF. These densities should be shown for the three maps obtained with EF. Without showing this appropriate supporting material, the quality of models shown Fig 2 cannot be assessed. The authors show the density corresponding to the N-terminal helix of LF (Fig S4), which is not novel nor relevant. Moreover, at the contour level shown Fig S4, the secondary structures of LF after aa 50 are not visible. This suggests low occupancy of this N-terminal helix compared to the rest of the molecule. Is this also the case for the N-terminal helix of EF? Are there any differences between the 3 maps obtained with EF? The model was build on the map containing only one LF molecule.

The authors describe that substrate "moved up" away from upon conversion from the pore conformation (Fig. 2). However, to reach this conclusion, the authors compare their model of PA-EF with that of their previously structure of PA-LFN. The comparison of EF with LFN is not appropriate to reach such conclusion. The comparison should be made between LFN-PA pore and LFN-PA prepore. Alternatively, the authors should determine the structure of PA-EF complex in the prepore conformation.

Others comments:

Do the authors plan to submit the EM map with associated models? Only the pbd reports were deposited.

More details about the purification of heptameric PA should be indicated, especially since the reference cited '9' (published in 2009) refers to other manuscripts published earlier, which refer to previous manuscripts

Reviewer #2 (Remarks to the Author):

Hardenbrook et al. have solved the cryoEM structures of PA heptamer and octamer forms bound to LF and AF. These structures represent the pre-channel and channel stages of the PA oligomer and the interaction of LF and EF with the α clamp. These studies provide a detailed view of the interaction of EF with pre-channel and channel PA (3 cryoEM structures) and a medium resolution of LF with PA. These studies show that the two amino terminal domains of EF and LF form α helices upon their interaction with the pre-channel state of PA, which shows how helices with different primary structures can be accommodated in the binding site. They further show that while EF undergoes a significant rearrangement of the helical and adenylate cyclase domains with respect to the PA binding domain upon its interaction with PA(7) whereas the same domains in LF are highly flexible and could not be imaged. There are some new aspects of the interaction of LF and EF with PA presented herein and some that confirm earlier observations with the cocrystal structure of LF and PA(8) previously solved by this group in 2010. Generally, this is an important study, but would have been more interesting if the investigators had leveraged some of the structural information to test specific aspects of their general model of the early steps of translocation.

1. Several of the figures (e.g., 1b, 1c, 2b and 2d) are quite complex (especially due to the number of tagged residues) and difficult to visualize. They would be less complex and provide more of an impact if they were presented as stereographic figures rather than 2D figures.
2. In regards to the displacement of Asps 184 and 87 from Lysines 213 and 214. Asp195 also forms a salt bridge with Lys214 (as well as Asp 184) and there appears to be movement of the loop during the pre-channel to channel states, which composed of the 2 nearby β strands (~residues 191-205) wherein Asp195 is present. Asp195 is also near H211, which has an estimated pKa of just under 6 (using ProPka). Is it possible that changes in the relationship between this triad of residues as the pH drops might be involved in the pH-triggered translocation event?
3. It is somewhat of a conundrum that EF appears to undergo a reorganization of its ACD and HD with respect to the PABD whereas the LF domains remain quite flexible with respect to its PABD. The authors suggest that the reorganization of the EF domains is important for its translocation, yet the same does not appear to be true for LF. It would have been useful to test whether or not the EF reorganization was necessary by mutational analysis of the salt bridges (and perhaps a few other residues at this interface), which appeared important in stabilizing this reorganization.
4. The yellow letters in fig. 3c are quite difficult to read and need to be changed to a different color.

Reviewer #3 (Remarks to the Author):

The manuscript entitled "Atomic structures of anthrax toxin protective antigen channels bound to partially unfolded lethal and edema factors" presents a number of high resolution cryoEM reconstructions of the two proteins in complex with the translocon, protective antigen (PA). The authors implement image processing strategies that include symmetry expansion and focused classification to overcome stoichiometric heterogeneity of complexes. In doing so, they are able to obtain atomic resolution details for both EF-PA and LF-PA complexes and derive a molecular mechanism for how protein translocation is coordinated across the alpha and phi clamps of PA. The

structural data they present is sound and the model building procedures/statistics are robust. The manuscript would however be further improved by augmenting the introduction and discussion sections.

1) Abstract: Here the authors describe the alpha clamp as a "deep amphipathic cleft". The figures in the main text and supplement give a detailed labelling of selected side-chain/side-chain interactions. The amphipathic nature of the cleft would be better demonstrated by a surface filled electrostatic potential representation, showing complementary interfaces between PA and EF/LF.

2) Abstract: Here the authors mention a "critical hairpin in PA". There is no mention of the word hairpin anywhere else in the text. I assume they are referring to Figures 2b and 1b, but it should be more obvious how they highlight this feature. Also, what evidence is there that this hairpin is "critical"? There are no functional/mutational studies done here. Are there appropriate literature references that could be included to support this statement in a revised introduction?

3) Introduction: It is unclear from the current introduction what the functional differences are between EF and LF? This should be expanded on in the introduction to make it more accessible to broader readership.

4) Introduction: If the "hairpin" in PA is critical for translocation, the current state of the literature should be included in the introduction to provide more context to the authors results.

5) Results: The authors determined 1 structure of a single LF in complex with PA, and 3 configurations of the EF/PA complex with 1 or 2 copies of EF bound. Is there a functional significance to these differences in stoichiometry? Is there any cooperativity of EF for its function? Is it sterically impossible to have more than one LF bound to PA?

6) Results p7, line 140: "This elevation in EF alpha1 appears to be caused by a change in the orientation of..." how do you know it is causation, not correlation? Are there any mutagenesis data in the literature that support the functional relevance of this?

7) Results and Supplementary figures: Which FSC curve is shown? Is it the mask-corrected FSC?

8) Methods section p. 27 line 411: "Heptameric PA was prepared as described⁹." It would be better to briefly describe the protocol and then cite the previous study for more detail.

9) Discussion: The discussion section is very brief. The manuscript would be improved to have this expanded to include a discussion of the significance of differences in the calmodulin-bound structure. It is not clear from the manuscript what the functional relevance of binding calmodulin is in this system; therefore, it is difficult to understand why the authors highlight the comparison in Figure 3.

10) Discussion: following on from point 9, the introduction mentions that anthrax toxin provides a useful model system for understanding protein translocation in general. I would like to see this expanded in the discussion section to understand what insight has been gained from cryoEM structures of other protein translocation systems, highlighting any differences with the anthrax system and common themes.

Reviewers' comments:

Reviewer #1 (Remarks to the Author):

In this manuscript, Hardenbrook and co-authors, presents two models of the anthrax toxin complexes with the protective antigen (PA) in a pore conformation: one bound to the lethal factor (LF) and the other to the edema factor (EF). The models were build using two cryo-EM maps, one with one N-terminal domain of LF (LFN) and the other with one EF molecule bound at the surface of the PA pore. The authors also obtained two other maps with EF that contain two EF molecules at different position at the surface of the PA heptameric pore (which they did not analyze).

Novelties of the work are the fact that PA is in a pore conformation and that this paper presents the first structure of PA bound to EF. Not surprisingly, given the similarity of the interacting surface between the PA oligomer in the pore and prepore conformations, LFN is seen interacting in a similar manner to that observed by X-ray crystallography and published by the same group (Feld et al, 2010), so this is not particularly novel. Most interesting is the structure of PA-EF, although as explained below, the authors should be careful in their interpretation.

The quality of cryo-EM work is of high standard and the use of symmetry expansion combined with masking for classification smartly done. However, the stoichiometry of binding obtained differs from previous report of the literature, where it has been clearly established that three substrates molecules can bind at the surface of PA heptamer. In this work, the authors pre-assembled and purified PA-LF and PA-EF complexes, then converted the PA to a pore conformation using urea (2M). The reduction in the number of substrates after urea treatment is highly concerning as is the unfolding of LFC. The concern here is that urea also unfolds LF (Lo et al, 2015).

Ans: The final concentration of urea is 1 M, as equal volumes of 2 M Urea and PA in buffer were added. This has been clarified in the methods. As per Lo et al, LF appears to remain fully folded and functional after 24 hours in 1 M urea. Our incubation was only for 5 minutes, allowing far less time for unfolding (see lines 589-590).

It is also to be noted that one remaining LFN (instead of three full-length molecules) is seen in 10% of the complexes (or after losing 97% of the bound substrate). What is the effect of urea on EF? Careful biochemical characterization of the substrates and the complexes should be conducted before being able to reach the conclusions made regarding the reorganization of the substrates upon binding to the PA channel ("EF domains reorganize upon binding the PA channel"). Could the domains reorganization of EF described Fig 3 be due to the urea treatment? A better approach would be to convert PA to a pore before assembling the complex, this way avoiding exposing LF and EF to urea.

Ans: We have not done experiments on the effects of urea on EF specifically. But we have recently obtained cryo-EM structures of LF bound to PA prechannel and EF bound to PA prechannel without urea treatment. The same exact domain reorganization of EF is seen in the prechannel complex, indicating that the 1 M urea treatment did not cause unfolding of the toxins.

An important concern as a reviewer is that the authors focus on the models and do not present the experimental evidence that support the models, especially regarding the position of N-terminal helix of EF at the surface of PA. The authors state that the N-terminal helix of EF

interacts with the α -clamp at the surface of PA. But the authors only show the model (Fig 2): the authors do not show the corresponding densities in any of three EM maps of PA-EF. These densities should be shown for the three maps obtained with EF. Without showing this appropriate supporting material, the quality of models shown Fig 2 cannot be assessed.

Ans: The requested density of the EF N-terminal helix is now shown in Figure S4. The quality of the helical density within this region is sufficient for atomic model building. For clarity, the requested densities in all three EM maps of PA-EF are shown below.

The authors show the density corresponding to the N-terminal helix of LF (Fig S4), which is not novel nor relevant. Moreover, at the contour level shown Fig S4, the secondary structures of LF after aa 50 are not visible. This suggests low occupancy of this N-terminal helix compared to the rest of the molecule. Is this also the case for the N-terminal helix of EF?

Ans: Our 3D classification results show that either one or none LF binds to one PA channel (Fig. S3a). Meanwhile, it is known that the N-terminal structure of LF has to **refold as an α helix** while binding within the α clamp. This evidence indicates that the occupancy of this N-terminal helix is good in the LF-bound structure. As with LF, the occupancy of the N-terminal helix in the EF complex is also good.

The authors describe that substrate “moved up” away from upon conversion from the pore conformation (Fig. 2). However, to reach this conclusion, the authors compare their model of

PA-EF with that of their previously structure of PA-LFN. The comparison of EF with LFN is not appropriate to reach such conclusion. The comparison should be made between LFN-PA pore and LFN-PA prepore. Alternatively, the authors should determine the structure of PA-EF complex in the prepore conformation.

Ans: Ideally, we should compare the structures of LFN-PA pore / LFN-PA prepore. Unfortunately, the 4.6 Å resolution structure of LFN-PA pore was not sufficient to see atomic details. However, the residues **involved in** destabilization are identically conserved between LF and EF (Fig. 2e), thus allowing us to generate this comparison as an alternative.

Others comments:

Do the authors plan to submit the EM map with associated models? Only the pdb reports were deposited.

Ans: Yes, the cryoEM maps with the associated models have both been submitted recently (see accession codes in the revised Table S1).

More details about the purification of heptameric PA should be indicated, especially since the reference cited '9' (published in 2009) refers to other manuscripts published earlier, which refer to previous manuscripts

Ans: We have added in more details about the purification of heptameric PA in the revised manuscript (see lines 568-573).

Reviewer #2 (Remarks to the Author):

Hardenbrook et al. have solved the cryoEM structures of PA heptamer and octamer forms bound to LF and AF. These structures represent the pre-channel and channel stages of the PA oligomer and the interaction of LF and EF with the α clamp. These studies provide a detailed view of the interaction of EF with pre-channel and channel PA (3 cryoEM structures) and a medium resolution of LF with PA. These studies show that the two amino terminal domains of EF and LF form α helices upon their interaction with the pre-channel state of PA, which shows how helices with different primary structures can be accommodated in the binding site. They further show that while EF undergoes a significant rearrangement of the helical and adenylate cyclase domains with respect to the PA binding domain upon its interaction with PA (7) whereas the same domains in LF are highly flexible and could not be imaged. There are some new aspects of the interaction of LF and EF with PA presented herein and some that confirm earlier observations with the cocrystal structure of LF and PA(8) previously solved by this group in 2010. Generally, this is an important study, but would have been more interesting if the investigators had leveraged some of the structural information to test specific aspects of their general model of the early steps of translocation.

1. Several of the figures (e.g., 1b, 1c, 2b and 2d) are quite complex (especially due to the number of tagged residues) and difficult to visualize. They would be less complex and provide more of an impact if they were presented as stereographic figures rather than 2D figures.

Ans: We thank the reviewer for pointing this out. We have relabeled these figures for better visualization. We tried stereographic panels at the beginning of figure preparation, but finally removed them because many young **readers don't know how to view** stereographic figures. We hope the supplemental movie will serve a similar purpose.

2. In regards to the displacement of Asps 184 and 87 from Lysines 213 and 214. Asp195 also forms a salt bridge with Lys214 (as well as Asp 184) and there appears to be movement of the loop during the pre-channel to channel states, which composed of the 2 nearby β strands (~residues 191-205) wherein Asp195 is present. Asp195 is also near H211, which has an estimated pKa of just under 6 (using ProPka). Is it possible that changes in the relationship between this triad of residues as the pH drops might be involved in the pH-triggered translocation event?

Ans: This may be true that a drop in pH could trigger a loosening of the binding interaction. We worked at neutral pH and found that mutating the residues in the salt bridges did not weaken the KD; therefore, the salt bridges were weakened due to the displacement of PA Lys 213 and Lys 214. This experiment has been added to the paper (see lines 212-215 and Fig. S6).

3. It is somewhat of a conundrum that EF appears to undergo a reorganization of its ACD and HD with respect to the PABD whereas the LF domains remain quite flexible with respect to its PABD. The authors suggest that the reorganization of the EF domains is important for its translocation, yet the same does not appear to be true for LF. It would have been useful to test whether or not the EF reorganization was necessary by mutational analysis of the salt bridges (and perhaps a few other residues at this interface), which appeared important in stabilizing this reorganization.

Ans: Following this suggestion, we made PABD mutations at the binding interface of EF (D171A and D174A) to test whether loss of salt bridges affects binding in the channel at neutral pH. As shown in new Fig. S6, there is no obvious change in binding affinity upon mutating these residues, indicating that these salt bridges are lost upon displacement of the PA loop containing Lys 213 and 214. This supports our conclusion that upon conversion to the channel conformation, this binding interface between PA and substrate is weakened (see lines 212-215 and Fig. S6).

4. The yellow letters in fig. 3c are quite difficult to read and need to be changed to a different color.

Ans: The yellow lettering in Fig. 3c has been changed to be more legible.

Reviewer #3 (Remarks to the Author):

The manuscript entitled “Atomic structures of anthrax toxin protective antigen channels bound to partially unfolded lethal and edema factors” presents a number of high resolution cryoEM reconstructions of the two proteins in complex with the translocon, protective antigen (PA). The authors implement image processing strategies that include symmetry expansion and focused classification to overcome stoichiometric heterogeneity of complexes. In doing so, they are able to obtain atomic resolution details for both EF-PA and LF-PA complexes and derive a molecular mechanism for how protein translocation is coordinated across the alpha and phi clamps of PA. The structural data they present is sound and the model building procedures/statistics are robust. The manuscript would however be further improved by augmenting the introduction and discussion sections.

1) Abstract: Here the authors describe the alpha clamp as a “deep amphipathic cleft”. The

figures in the main text and supplement give a detailed labelling of selected side-chain/side-chain interactions. The amphipathic nature of the cleft would be better demonstrated by a surface filled electrostatic potential representation, showing complementary interfaces between PA and EF/LF.

Ans: Initially we considered using electrostatic potential surface, while we also want to distinguish two neighboring PA subunits, thus we finally choose transparent surface combined with colored PA ribbon for representation. The amphipathic nature of the cleft has been described by Feld et al 2010. Here we show electrostatic potential surface renderings of the PA channel for both the (a) PA-LF and (b) PA-EF complexes:

Electrostatic surface representations of the amphipathic α -clamp cleft in the PA channel for both the PA-LF and PA-EF complexes. The (a) LF and (b) EF α 1 helices are shown as ribbons. The PA₇ channel is shown as an electrostatic surface.

2) Abstract: Here the authors mention a “critical hairpin in PA”. There is no mention of the word hairpin anywhere else in the text. I assume they are referring to Figures 2b and 1b, but it should be more obvious how they highlight this feature. Also, what evidence is there that this hairpin is “critical”? There are no functional/mutational studies done here. Are there appropriate literature references that could be included to support this statement in a revised introduction?

Ans: Good suggestions. We changed the term “**hairpin**” to “**loop**”. Lacy et al. 2005 showed the importance of this loop in LF_N binding, where charge reversal mutations could complement the binding affinity. Additionally, Feld et al. 2010 showed the salt bridges in a 3.2 Å crystal structure of the PA8 prechannel-LF_N4 complex. This literature was cited in the revised manuscript (see lines 62-68).

3) Introduction: It is unclear from the current introduction what the functional differences are between EF and LF? This should be expanded on in the introduction to make it more accessible to broader readership.

Ans: The functions of LF and EF were added to the introduction of the manuscript to clarify their functional differences (see lines 33-45).

4) Introduction: If the “hairpin” in PA is critical for translocation, the current state of the literature should be included in the introduction to provide more context to the authors results.

Ans: We have changed “hairpin” to “loop” and referred to Feld et al. 2010 and Lacy et al. 2005 regarding the critical function of this loop in LF_N binding (see lines 62-68).

5) Results: The authors determined 1 structure of a single LF in complex with PA, and 3 configurations of the EF/PA complex with 1 or 2 copies of EF bound. 1) Is there a functional significance to these differences in stoichiometry? Is there any cooperativity of EF for its function? 2) Is it sterically impossible to have more than one LF bound to PA?

Ans: 1) There is no identified functional significance to different stoichiometry, and no known cooperativity for EF function. There is a cooperativity for binding of a small peptide to the channel, however.

2) Yes. Ideally, PA₇ can bind up to 3 LF or 3 EF. As the binding interface appears to be destabilized, the additional LF and EF appear to dissociate during the purification of the nanodisc complexes.

6) Results p7, line 140: “This elevation in EF alpha1 appears to be caused by a change in the orientation of...” how do you know it is causation, not correlation? Are there any mutagenesis data in the literature that support the functional relevance of this?

Ans: We have changed “**caused by**” to “**related to**” for toning down the inference (see line 183).

7) Results and Supplementary figures: Which FSC curve is shown? Is it the mask-corrected FSC?

Ans: Yes, the mask-corrected FSC curves are shown in the paper (see lines 710-712).

8) Methods section p. 27 line 411: “Heptameric PA was prepared as described⁹.” It would be better to briefly describe the protocol and then cite the previous study for more detail.

Ans: The complete protocol has been added to the revised manuscript (see lines 568-573).

9) Discussion: The discussion section is very brief. The manuscript would be improved to have this expanded to include a discussion of the significance of differences in the calmodulin-bound structure. It is not clear from the manuscript what the functional relevance of binding calmodulin is in this system; therefore, it is difficult to understand why the authors highlight the comparison in Figure 3.

Ans: EF function is expanded on in the introduction, giving insight into the relevance of CaM for its function. Currently the only full-length structure available of EF containing all domains is EF bound to CaM, therefore making this the only structure available to compare it to. Compared to this structure, EF takes on a different domain organization.

10) Discussion: following on from point 9, the introduction mentions that anthrax toxin provides a useful model system for understanding protein translocation in general. I would like to see this

expanded in the discussion section to understand what insight has been gained from cryoEM structures of other protein translocation systems, highlighting any differences with the anthrax system and common themes.

Ans: Thanks for the suggestion. The revised manuscript has been edited to include other protein translocation systems (see lines 303-319).

Table S1. CryoEM data collection and refinement statistics.

Data collection	LF complex	EF complex		
	PA ₇ -LF	PA ₇ -EF	PA ₇ -1,3-EF	PA ₇ -1,4-EF
EM equipment	FEI Titan Krios		FEI Titan Krios	
Voltage (KV)	300		300	
Detector	Gatan K2		Gatan K2	
Pixel size (Å)	1.07		1.07	
Electron dose (e ⁻ /Å ²)	62.9		60.2	
Defocus range (µm)	-1.5 ~ -3.0		1.5 ~ -3.0	
Reconstruction				
Software	RELION2.1		RELION2.1	
Number of used particles	63,807	333,455	72,864	73,784
Accuracy of rotation (°)	2.51	1.62	1.36	1.41
Accuracy of translation (pixels)	1.70	0.76	0.71	0.72
Map sharpening B-factors (Å ²)	-169.7	-80	-80	-80
Resolution FSC 0.143 (Å)	4.6	3.2	3.4	3.4
Model building				
Software	COOT	COOT	COOT	COOT
Refinement				
Software	PHENIX	PHENIX	PHENIX	PHENIX
Resolution (Å)	4.6	3.2	3.4	3.4
R-factor	0.37			
Number of protein residues	4,154	3,681	4,401	4,401
Map CC	0.81	0.86	0.84	0.85
R.m.s deviations				
Bonds length (Å)	0.01	0.01	0.01	0.01
Bonds angle (°)	1.15	1.17	0.90	0.94
Ramachandran plot statistics (%)				
Preferred	96.67	95.49	93.30	92.72
Allowed	3.33	4.51	6.69	7.18
Outlier	0	0	0	0.1
Rotamers outliers (%)	0.40	0.29	0.21	0.35
C-beta deviations	0	0	0	0
Clash score	6.78	4.83	5.34	5.24
MolProbity score	1.59	1.57	1.72	1.74
PDB code	6PSN	6UZB	6UZD	6UZE
EMDB code	EMD-20459	EMD-20955	EMD-20957	EMD-20958

REVIEWERS' COMMENTS:

Reviewer #2 (Remarks to the Author):

The authors have answered the comments adequately with additional explanations, text and experiments. I do not see any additional problems with this work, which is quite well done, expansive and provides new atomic level insights into a complex translocation mechanism that has broad importance to the field of A-B toxin translocation mechanisms.